# OpenReview forum: "VERA-V: Variational Inference Framework for Jailbreaking Vision-Language Models"
_ICML.cc/2026/Conference — ICML 2026 regular_

### Official Review · Reviewer_wuzP · 2026-03-10

**Soundness:** 4
**Presentation:** 4
**Significance:** 3
**Originality:** 3
**Overall Recommendation:** 5
**Confidence:** 4

**Summary:**

This paper proposes VERA-V, a variational inference framework for multimodal red-teaming that learns a joint distribution over paired text–image prompts to jailbreak vision–language models (VLMs). An attacker LLM parameterized via LoRA generates a coupled pair of latent prompts, which are transformed into a composite visual input using typographic rendering, diffusion-based image synthesis, and structured distractors; the attacker is optimized with a REINFORCE estimator using a judge model as a proxy likelihood signal. Experiments on HarmBench and HADES across open-source and commercial VLMs report large gains in attack success rate (ASR) over prior multimodal methods and show claims of scalability, diversity, and transferability.

**Compliance With Llm Reviewing Policy:**

Affirmed.

**Final Justification:**

I am keeping my score at 5 because the paper presents a strong and timely contribution, and the rebuttal further clarified the novelty, strengthened the empirical evidence, and addressed my main concerns sufficiently for me to remain positive about acceptance.

**Key Questions For Authors:**

### Q1. Necessity of joint multimodal optimization
The proposed framework jointly optimizes text and image prompts. However, several components used in the pipeline (e.g., typography-based attacks, diffusion-generated images, and distractor images) have been explored in prior work. Could the authors provide additional analysis or ablation studies comparing:
- text-only optimization,
- image-only optimization, and
- joint text–image optimization?
Such comparisons would help clarify whether the main performance gains come from the proposed joint optimization strategy or from the combination of existing attack techniques.

### Q2. Relationship between the judge model and the evaluation protocol
The attacker is optimized using the HarmBench validation classifier as the judge model, while the evaluation also relies on HarmBench-related classifiers and LLM-based judges. This raises a potential concern regarding the relationship between the training judge and the evaluation procedure.
Could the authors clarify:
- To what extent the training judge model overlaps with the evaluation signals used to compute ASR?
- Whether the reported results could be influenced by the attacker implicitly optimizing for the evaluation classifier or related scoring models, rather than exploiting genuine vulnerabilities in the target VLM?
It would be helpful if the authors could further discuss or experimentally analyze the separation between the judge used for optimization and the models used for evaluation, for example by evaluating attacks with judges that are independent of the HarmBench evaluation classifiers. This would help verify that the attack is targeting the VLM itself rather than the evaluation pipeline.

### Q3. Generalization beyond the current attack components
The current framework combines several existing techniques (typography rendering, diffusion image synthesis, distractor images).
Could the authors elaborate on whether the VERA-V framework is agnostic to the specific attack primitives used in the composite image? For example:
- Can the framework incorporate other types of visual perturbations or multimodal attack strategies?
- Would the optimization process remain effective if some of the current components (e.g., distractor images or typography rendering) were removed or replaced?
Understanding the flexibility of the framework would help clarify whether the proposed approach represents a general attack framework or primarily a specific combination of existing techniques.

**Limitations:**

yes

**Strengths And Weaknesses:**

### Strengths
（1）Important problem
The paper addresses multimodal jailbreak attacks against VLMs, which is an increasingly important research direction as vision-language models become widely deployed in real-world systems. Studying vulnerabilities in multimodal settings is timely and valuable for improving model safety.

（2）A unified framework for multimodal jailbreak discovery
The proposed framework introduces a joint optimization of text and image prompts, enabling coordinated adversarial signals across modalities. Modeling prompt generation as a distribution over paired prompts allows the system to produce multiple attack variants rather than single handcrafted instances.

（3）Clear and modular attack pipeline
The overall pipeline is well structured and easy to understand:
- attacker LLM generates prompt pairs
- typography rendering and diffusion image synthesis
- distractor images
- composite adversarial image
- judge-based feedback optimization
This modular design makes the framework relatively extensible.

（4）Empirical results on multiple models
The experimental evaluation covers both open-source and closed-source VLMs, including GPT-4o. The reported improvements in ASR over prior baselines are substantial, suggesting that the attack strategy is effective in practice.

### weakness
(1) The contribution mainly lies in system integration
Many key components of the attack pipeline appear to be adapted from prior work, including:
- typography-based attacks
- diffusion-generated adversarial images
- distractor images
- reinforcement-learning-based prompt search
The primary contribution seems to be combining these techniques with joint prompt optimization rather than introducing fundamentally new attack mechanisms.

(2) Potential evaluation bias from the judge model
The attacker is trained using a judge model (HarmBench classifier) that assigns harmfulness scores. However, evaluation also relies on HarmBench-related classifiers and LLM judges.
This raises a potential concern that the attack may partially optimize for the judge model rather than the target VLM, which could inflate ASR results.
A clearer separation between training and evaluation signals would strengthen the experimental validity.

---

> ### Author Rebuttal · Authors · 2026-03-31
>
> # Response to Reviewer wuzP
>
> We appreciate the reviewer for raising these concerns and welcome the opportunity to clarify them.
>
> ---
>
> ## Q1 & W1: Novel Attack Mechanism and Joint Optimization
>
> VERA-V casts multimodal jailbreaking as posterior inference over paired text-image prompts, **providing the key insight that effective VLM attacks require correlated explicit and implicit signals** describing the same harmful intent. Prior methods treat modalities independently. VERA-V's joint posterior $q_\theta(x_t, x_v)$ learns these correlations automatically through optimization. The new attack mechanism we introduce is the joint optimization of cross-modal attacks, which is non-trivial to achieve. We showed that text-only optimization attackers perform poorly when applied to VLMs (**Appendix H**). The components (typography, diffusion, distractors) become effective and stealthy when coordinated through VERA-V's jointly learned distribution.
>
> To further evaluate multimodal optimization effectiveness, we conduct additional ablation studies: our proposed joint text-image optimization framework (VERA-V) achieves the highest ASR at 80%. When isolating the modalities, text-only optimization yields a 60% ASR, while diffusion-based image-only optimization achieves merely 12%. Notably, joint text-image optimization (without distractors) reaches a 65% ASR. This demonstrates that joint multimodal optimization outperforms single-modality attacks, and the further introduction of visual distractors in VERA-V is critical for maximizing attack robustness. All variants are optimized using the VERA-V framework except for the composition methods.
>
> | Composition Method | ASR |
> |--------------------|-----|
> | Text-only optimization | 60% |
> | Diffusion-based image-only | 12% |
> | Distractors only | 0% |
> | Joint text-image optimization (no distractor images) | 65% |
> | Joint text-image optimization (VERA-V) | 80% |
>
> ---
>
> ## Q2 & W2: Evaluation Protocol and Judge Model
>
> We clarify that we follow the protocol recommended by HarmBench [1] to prevent evaluation overfitting. HarmBench explicitly provides two distinct classifiers for this purpose:
>
> - **Validation classifier (Mistral-7B):** intended for use during the jailbreak attack.
> - **Test classifier (LLaMA2-13B):** intended for final evaluation.
>
> We use the validation classifier as our judge during VERA-V optimization and the test classifier for ASR evaluation (similar to [2, 3]).
>
> We also report ASR using GPT-4o-mini as an independent evaluator (Tables 1-2), which has no relationship to our training judge. VERA-V achieves consistently strong performance across both evaluation models (e.g., 72.0% with HarmBench test classifier vs. 71.0% with GPT-4o-mini on Qwen2.5-VL), demonstrating that our attacks exploit genuine VLM vulnerabilities rather than overfitting to the judge.
>
> **Training Judge Ablation:** **Appendix E.5 (Table 9)** further shows VERA-V performs well across different judge models for training (HarmBench validation: 80%, GPT-4o-mini: 76%, StrongReject: 72%, and using HarmBench test classifier as evaluation classifier), confirming robustness to judge choice.
>
> ---
>
> ## Q3.1: Framework Generalizability to Other Attack Primitives
>
> Yes, VERA-V is agnostic to specific attack primitives. Its joint optimization framework for cross-modal attacks can be applied to other types of strategies. In this work, we primarily demonstrate how to utilize different cues (typography, diffusion, distractors) to achieve effective attacks and show that joint optimization is key to their effectiveness. However, the framework can incorporate any perturbation or transformation as long as it can be embedded into the composite input and the judge model provides meaningful feedback for optimization.
>
> ---
>
> ## Q3.2: Component Removal Flexibility
>
> Components can be removed as our ablations demonstrate in **Appendix E.1**: performance changes but the framework remains applicable. Without distractors (65% ASR), without diffusion (60% ASR), VERA-V continues to function, confirming its robustness to different component configurations. This flexibility establishes VERA-V as a general optimization framework for multimodal adversarial discovery.
>
> ---
>
> ## References
>
> [1] Mazeika, M., Phan, L., Yin, X., Zou, A., Wang, Z., Mu, N., ... & Hendrycks, D. (2024). Harmbench: A standardized evaluation framework for automated red teaming and robust refusal.
>
> [2] Lochab, A., Yan, L., Pynadath, P., Zhang, X., & Zhang, R. (2025). Vera: Variational inference framework for jailbreaking large language models.
>
> [3] Liu, H., Xie, Y., Wang, Y., & Shieh, M. (2024). Advancing adversarial suffix transfer learning on aligned large language models.

---

> > ### Author Rebuttal · Reviewer_wuzP · 2026-04-03
> >
> > Thank you for the rebuttal. The added ablations and clarification on the judge/evaluation protocol are helpful and address most of my concerns. In particular, the new results support the claim that joint text-image optimization contributes beyond single-modality variants, and the separation between the HarmBench validation classifier for optimization and the HarmBench test classifier / GPT-4o-mini for evaluation makes the evaluation more convincing.

---

> > > ### Author Response · Authors · 2026-04-03
> > >
> > > # Final response to Reviewer wuzP
> > >
> > > We express our gratitude to the reviewer again for your valuable time and constructive feedback!

---

### Official Review · Reviewer_PuSW · 2026-03-12

**Soundness:** 2
**Presentation:** 2
**Significance:** 2
**Originality:** 2
**Overall Recommendation:** 3
**Confidence:** 3

**Summary:**

This paper proposes VERA-V, a multimodal jailbreak framework. The authors intend to improve the diversity and effectiveness of multimodal jailbreak attacks beyond fixed prompt templates. Overall, the study considers a broad theme by combining typography-based harmful text, diffusion-generated images, and distractor images into a joint optimization framework.

**Compliance With Llm Reviewing Policy:**

Affirmed.

**Final Justification:**

The rebuttal clarifies several important points, especially the construction of the composite image and the intended interpretation of Eq.(3), which improves my confidence compared with the original submission. However, the paper still has notable clarity issues: several formulations are difficult to follow when first introduced, and the connection between components is not always well motivated. The novelty beyond combining existing multimodal jailbreak ingredients also remains moderate. Overall, the rebuttal improves my assessment slightly, so I raise my score but keep a weak reject recommendation.

**Key Questions For Authors:**

1. Could the authors provide a precise algorithmic description of how the composite image ($v_{comp}$) is constructed, including layout, resolution, and ordering of components?
2. Can the authors isolate the contribution of typography, diffusion images, and distractor images through stronger ablation studies?
3. Several equations and notation choices appear imprecise.

**Limitations:**

Yes

**Strengths And Weaknesses:**

Strengths

1. The attempt to jointly optimize textual and visual latent prompts is interesting and may offer a broader search space than fixed attack templates.
2. The paper compares against several recent multimodal jailbreak baselines.

Weaknesses

1. Several mathematical formulations are imprecise, which weakens confidence in the technical rigor. For example, Eq. (3) formulates the jailbreak objective as $((x,v)\sim P_{VLM}((x,v)\mid y\in Y_h))$, whereas the probabilistic quantity should more naturally be expressed as the probability of harmful output conditioned on input, e.g. $(P_{VLM}(y\in\mathcal{Y}_h \mid (x,v)))$.
2. The method description lacks important implementation details. In particular, the construction of the composite image $(v_{comp})$ is insufficiently specified.
3. The overall presentation is difficult to follow. Several sections introduce concepts before clearly defining them, and the logical connection between preliminaries, formulation, and optimization is often unclear.
4. The novelty relative to prior multimodal jailbreak methods is not sufficiently justified. The framework mainly combines existing ingredients (typography, diffusion generation, distractors) and it remains unclear whether the gain comes from the joint optimization itself or from specific prompt engineering choices.

---

> ### Author Rebuttal · Authors · 2026-03-31
>
> # Response to Reviewer PuSW
>
> We thank the reviewer for the thoughtful feedback and the opportunity to clarify these points.
>
>
> ## W1:
>
> Please check Section Q2 in our rebuttal to Reviewer eKww.
>
> ## W2 & Q1:
>
> We provide the strategy to build the composite image in Appendix A and Section 4.1.3. We provide examples of the final input to target models in Appendix L and offer precise specifications below:
>
> Layout:
> - **Structure:** 3×3 grid (9 cells total)
> - **Cell dimensions:** 300×300px image + 300×30px label area = 300×330px per cell
> - **Final resolution:** 900×990px (3 columns × 300px wide, 3 rows × 330px tall)
>
> Components:
> - 7 distractor images (cells 1-7)
> - 1 typographic image $v_T$ (cell 8)
> - 1 diffusion image $v_D$ (cell 9)
>
> **Ordering of components:** Distractors occupy top-left positions; adversarial images ($v_T$, $v_D$) are placed in the bottom-right to align with the fixed prompt $x_f$ referencing "pictures 8 and 9" (see Figure 7).
>
>
> ## W3:
>
> We note that the paper follows a logical structure, introducing concepts before they are used. Section 3 (Preliminaries) defines all foundational components: diffusion-based image generation (Eq. 1), typography transformation (Eq. 2), and the visual distraction strategy. Section 4 builds on these definitions: Section 4.1 formulates the jailbreak objective using these components, Section 4.2 derives the variational objective and optimization procedure, and Section 4.3 presents the complete algorithm. Figure 1 further provides a visual overview contrasting VERA-V with prior single-attack methods. We will add a brief roadmap at the end of Section 1 to make this flow more explicit. Should anything remain unclear, we would welcome concrete examples so that we can address them directly.
>
>
> ## W4: Novelty
>
> We clarify the novelty of VERA-V in regards to prompt engineering, joint optimization, and component synergy:
>
> **Prompt Engineering vs. Joint Optimization:** Prompt engineering involves manually crafting static templates or heuristics. We explicitly clarify that the initial text prompts used in our experiments are fundamental and static, lacking any sophisticated prompt engineering. We also provide the attacker system prompt in Appendix C and prompt template for VLM in Appendix J. Appendix E.2 (Table 6) directly isolates this: Best-of-N uses identical components but no posterior learning. VERA-V's 8× improvement over Best-of-N shows that joint optimization, not prompt engineering, drives performance.
>
> **Critical Role of Joint Optimization:** Single-modality optimization fails for VLMs. Appendix H (Table 10) shows text-only optimization: attacker achieves only 34.5% ASR vs. VERA-V's 72%. VLMs require coordinated cross-modal signals: the attacker must learn that $x_t$ (explicit instruction) and $x_v$ (implicit context) describe the same harmful intent from complementary perspectives. Without joint optimization, these signals remain uncorrelated and ineffective. Further, prior multimodal methods (e.g., FigStep, CS-DJ) largely rely on fixed templates and single-step heuristics. VERA-V introduces a fundamental paradigm shift for VLM red-teaming: joint refinement and scalable sampling of diverse, evasive attacks that prior static methods cannot achieve.
>
> **Learning Component Synergy:** Individual components fail because they lack coordination (Appendix E.1, W1&Q3 Reviewer 1): Diffusion-only (12% ASR) provides weak implicit cues; typography-only (56-70% ASR & ~34% Toxicity rate) triggers high toxicity detection. They become effective and elusive (80% ASR & 27% Toxicity rate) when VERA-V's posterior learning discovers how to coordinate them: typography provides explicit instruction, diffusion adds implicit reinforcement. When these modalities are combined and optimized independently, the attacks fail (see detailed analysis in response to Question 2). This coordination emerges from joint optimization—as shown above, static aggregation (Best-of-N) fails entirely.
>
>
> ## Q2:
>
> Our ablation studies in Appendix E.1 (Table 5) isolate the effect of the composite image design by keeping the VERA-V refinement pipeline fixed and modifying only the input composition. Please also check Section W1 in our rebuttal to Reviewer eKww, where we provide additional configurations and key insights of the ablation studies.
>
> Further, Appendix E.2 (Table 6) validates VERA-V's refinement mechanism improves performance by roughly ~8× over Best-of-N (which performs no posterior learning), establishing that both the composite design and joint optimization are essential.
>
>
> ## Q3:
>
> We thank the reviewer for pointing out the problem, but could the reviewer briefly explain which parts of the equations and notation choices are unclear? We would provide a detailed explanation in the future reply. We have clarified Eq.(3)'s form above.

---

> > ### Author Rebuttal · Reviewer_PuSW · 2026-04-02
> >
> > The rebuttal clarifies several implementation details and partially addresses my concerns, but some issues remain. In particular, the formulation in Section 4.1.1 is still not sufficiently clear from the paper itself: although Eq.(3) is intended as a posterior inference view, this motivation is not explained when the objective is first introduced, so readers may naturally interpret it as an imprecise probabilistic objective. More broadly, the paper remains difficult to follow because several components are introduced before their roles are clearly motivated. For example, Section 3 presents three preliminaries with mathematical definitions, but their relation to the proposed framework is not immediately clear, and in Eq.(5) the function Combine is introduced without specifying layout, ordering, or whether placement affects the attack behavior. I do not deny the empirical effectiveness of the method, but the technical presentation still needs improvement.

---

> > > ### Author Response · Authors · 2026-04-03
> > >
> > > # Final response to Reviewer PuSW
> > >
> > > We thank the reviewer for the continued discussion and for recognizing the empirical effectiveness of our method. We address the remaining presentation concerns below:
> > > ## Motivation for the Posterior Formulation (Eq. 3)
> > > The framing of jailbreak generation as a posterior inference problem is explicitly introduced multiple times before Section 4.1.1—specifically in the Abstract, the Introduction, and the opening paragraph of Section 4.
> > > However, we understand that the text immediately preceding Eq. (3) might momentarily read like a deterministic optimization goal. To prevent any localized ambiguity and better bridge this conceptual gap, we will revise the paragraph right before Eq. (3) in Section 4.1.1. We will add more explicit context regarding the posterior inference perspective at that specific location to ensure a smoother and clearer mathematical transition for the readers.
> > > ## Flow of the Preliminaries (Section 3)
> > > The Preliminaries section is intended to provide the mathematical definitions of the tools (Diffusion, Typography, Distractors) that are later jointly optimized in our framework. To improve readability, we will add a brief introductory sentence at the start of Section 3 to explicitly signpost that these three components serve as the building blocks for the explicit and implicit use in our method.
> > > ## Details of the Combine function (Eq. 5)
> > > We will include more implementation details of the Combine function in the final version. We will specify the grid layout, spatial ordering of the typography/diffusion/distractor images, and clarify how placement variations were handled during our evaluation.

---

### Official Review · Reviewer_6WMu · 2026-03-13

**Soundness:** 3
**Presentation:** 3
**Significance:** 3
**Originality:** 2
**Overall Recommendation:** 3
**Confidence:** 3

**Summary:**

This paper proposes VERA-V, a multimodal jailbreak framework for vision-language models that formulates attack generation as variational inference over paired text-image prompts. The key idea is to learn a joint attacker distribution over a typographic text prompt and a diffusion-based image prompt, then transform them into a composite visual input with distractor images and optimize the attacker using judge-model feedback through a REINFORCE-style objective. The paper argues that this joint probabilistic view captures cross-modal interactions better than fixed-template or single-shot attacks. Empirical results show large gains over baselines.

**Compliance With Llm Reviewing Policy:**

Affirmed.

**Final Justification:**

While some concerns have been partially addressed, I maintain my original score.  Even though the authors state that Ideator [1], Defense2Attack [2], and JPRO [3] do not release their code, they can still reimplement some of them by themselves. Additionally, I remain unconvinced of the novelty. The proposed approach appears to be an incremental combination of existing techniques, and the rebuttal did not provide sufficient evidence to address this concern. Therefore, I maintain my original score.

**Key Questions For Authors:**

1. What is the exact form of the prior in the ELBO, and how sensitive are results to this choice?
2. What is the total query budget per harmful behavior for VERA-V versus each baseline? Please report matched-budget comparisons in terms of target-model calls, judge calls, and wall-clock time.
3. How much of the final improvement comes from joint posterior learning itself versus the composite attack design? A strict ablation should compare: typography only, diffusion only, distractors only, pairwise combinations, and then full VERA-V under matched budgets.

**Limitations:**

yes

**Strengths And Weaknesses:**

Strengths:

1. Recasting multimodal jailbreak generation as joint posterior learning over text-image pairs is a more principled framing than the usual template engineering. It gives the paper a clearer conceptual identity than many attack papers that are mostly heuristic compositions.
2. On both HarmBench and HADES, VERA-V substantially outperforms prior baselines.
3. The paper includes transferability, defense-aware evaluation, and diversity/scalability discussion.

Weaknesses:

1. The paper does not convincingly isolate how much of the gain comes from the new learning framework versus simply combining three strong ingredients: typography, diffusion, and distractors.
2. The paper argues that direct comparison on diversity/scalability against prior multimodal methods is infeasible because prior methods are one-shot or template-based.
3. The paper itself states that VERA-V extends prior VERA from LLMs to the multimodal setting. That extension may still be valuable, but the paper needs to work harder to show that the multimodal jump is not merely VERA + known VLM attack components.

---

> ### Author Rebuttal · Authors · 2026-03-31
>
> # Response to Reviewer 6WMu
> We thank the reviewer for the insightful comments.
>
> ## W1 & Q3:
> To isolate the variational framework contribution, we compare against the Best-of-N (Appendix E.2), which uses identical components (typography + diffusion + distractors) but disables LoRA updates (no posterior learning) and samples N=40 ($N = S \times B$ prompts). Table 6 shows VERA-V (66% ASR) significantly outperforms Best-of-N (8% ASR), confirming gains stem from learned distributional structure of multimodal vulnerabilities, not composite design.
>
> We ablate each component to isolate the effect of each part on ASR (Appendix E.1) and time consumption under matched query budgets (50-subset of HarmBench, Qwen2.5-VL target).  Please refer to the table in section W1 in our rebuttal to Reviewer eKww.
>
> Results show: (1) distractors alone are ineffective (0% ASR); (2) diffusion alone is weak (12%) but improves with distractors (34%); (3) typography is the primary driver (60%) and benefits from diffusion (65%); (4)  the full VERA-V composition achieves the highest ASR (80%). This demonstrates VERA-V's refinement mechanism learns to exploit cross-modal synergies, combining modalities improves ASR without scaling time linearly.
>
> ## W2:
> Prior methods are inherently one-shot: they produce a single fixed attack per behavior with no learned distribution to sample from, limiting direct diversity/scalability comparison. However, we realized CS-DJ [1] can produce varied sub-queries by sampling with a higher temperature (0.7). To fairly evaluate, we conduct a time-matched repeated-trial experiment allocating CS-DJ the same compute time budget as VERA-V, allowing CS-DJ repeated iterations:
>
> **Scalability**: CS-DJ plateaus at 42.4% ASR vs VERA-V's 80%. CS-DJ performs repeated independent trials with diminishing returns, while VERA-V amortizes optimization into a learned distribution, enabling highly successful attacks after a short training.
>
> **Diversity**: CS-DJ shows 0.465 Self-BLEU and 0.105 BLEU-to-template, vs VERA-V’s 0.225 and 0.0001. Lower BLEU indicates higher diversity. VERA-V generates diverse attacks probing different vulnerability modes.
>
> ## W3:
>
> VERA-V is not a trivial combination of existing components, but a fundamental restructuring to exploit cross-modal interactions. Extending a text-only framework to multimodal is not straightforward. Generating cross-modal attacks introduces several non-trivial methodological challenges:
>
> 1. **Coupled Latent Space Optimization**: While VERA optimizes a single text prompt, VERA-V must learn a joint posterior over two correlated variables: $x_t$ (for typography) and $x_v$ (for diffusion), requiring a redesigned multi-stage rendering pipeline and objective to capture cross-modal interactions.
> 2. **Naive Text-Only Optimization Fails**: VLM safety relies on both textual semantics and visual attention patterns. Text-only methods fail entirely:  achieving only 34.87% ASR (Appendix H), demonstrating that text-only assumptions do not easily transfer. Thus, the gap between LLM and VLM jailbreak performance reflects multimodal difficulty, not triviality. Further, Appendix E.2 shows that using VERA-V components without refinement (Best-of-N) achieves only 8% ASR vs. VERA-V's 66%, demonstrating the need for joint optimization.
> 3. **Emergent Cross-Modal Synergies**: Individual components fail because they lack coordination (Appendix E.1): Diffusion-only (12% ASR) provides weak implicit cues; typography-only (56-70% ASR) triggers high toxicity. They become effective and elusive when VERA-V's joint posterior learning discovers how to coordinate explicit instruction and implicit reinforcement.
>
> ## Q1:
> Please see Section Q1 in our rebuttal to Reviewer eKww.
>
> ## Q2:
> Baselines use single-step attack strategies (1 target + 1 judge call per attack); CS-DJ additionally invokes a decomposer (3 calls) . FigStep and Hades use no attacker model. Detailed breakdown of model calls and wall-clock time at: https://anonymous.4open.science/r/72d5a1f707f86cd5450faf5ad3191663199aad3bd05fe81b/
>
> VERA-V's training is a one-time cost that amortizes across attack generation, enabling the sampling of new prompts at inference. As shown in Figure 2b, after a brief optimization phase (~100s), VERA-V samples diverse attacks, achieving high cumulative ASR within 400s. We expand this to a time-matched comparison (Response W2): VERA-V achieves 80% ASR versus CS-DJ's 42.4%.
>
> While VERA-V has higher per-query latency (4.0s vs 2.3s), it delivers high return per unit time, ~2× the ASR under identical compute. This produces significantly more diverse attacks (Self-BLEU 0.225 vs 0.465) that more effectively probe distinct vulnerability modes, whereas CS-DJ’s query decomposition yields structurally similar variants.
>
> Our experiments use batch size 4; increasing batch size improves throughput by parallelizing sample generation.
>
> [1] Yang et al. (2025). Distraction is all you need for multimodal large language model jailbreaking.

---

> > ### Author Rebuttal · Reviewer_6WMu · 2026-04-03
> >
> > Thank you for the detailed rebuttal. However, I still do not think the main issue is resolved. Since typography alone already achieves strong ASR, the current evidence still leaves open whether VERA-V exploits cross-modal coupling or mainly behaves as a stronger optimized typography attack with auxiliary visual components. A comparison against non-variational trainable multimodal baselines would be needed to support the paper’s central claim.

---

> > > ### Author Response · Authors · 2026-04-05
> > >
> > > # Final Reply to Reviewer 6WMu
> > >
> > > We thank the reviewer for the continued discussion. We believe our existing ablation study directly addresses this concern and demonstrates true cross-modal coupling.
> > >
> > > Consider the composition of the "Typography + Distractors" variant (70% ASR) versus the full "VERA-V" composition (80% ASR):
> > >
> > > - Typography + Distractors: The VLM receives a grid containing 8 distractor images, followed by 1 typographic image (placed at position 9).
> > > - VERA-V (Typography + Diffusion + Distractors): The VLM receives 7 distractor images, 1 jointly-optimized diffusion image, and 1 typographic image (placed at position 9).
> > >
> > > The typographic component is identical in both setups. By swapping out just one semantic-free distractor image for our jointly-optimized diffusion image, the attack success rate jumps by 10 percentage points. If VERA-V were primarily behaving as a "stronger optimized typography attack," the introduction of the diffusion image would not drive this synergistic gain. The 10% improvement occurs because our framework jointly couples the explicit text prompt with the implicit adversarial cues of the diffusion image.
> > >
> > > Regarding non-variational trainable multimodal baselines, we agree it would be a valuable comparison. However, optimizing attack components is nontrivial — the variational formulation is precisely our proposed solution to this optimization challenge. There is no straightforward non-variational alternative, and to our knowledge, no such baseline existed in the literature at the time of submission. We surveyed recent multimodal jailbreak methods that incorporate automated refinement mechanisms, such as Ideator [1], Defense2Attack [2], and JPRO [3], but none provide publicly available code. We will include a discussion of these concurrent methods and their relationship to our approach in the revised manuscript.
> > >
> > > [1] Wang, R., Li, J., Wang, Y., Wang, B., Wang, X., Teng, Y., ... & Jiang, Y. G. (2025). Ideator: Jailbreaking and benchmarking large vision-language models using themselves. ICCV.
> > >
> > > [2] Zhao, Y., Zheng, X., & Ma, X. (2025). Defense-to-Attack: Bypassing Weak Defenses Enables Stronger Jailbreaks in Vision-Language Models.
> > >
> > > [3] Zhou, Y., Bai, Y., Gao, K., Dai, T., & Xia, S. T. (2025). JPRO: Automated Multimodal Jailbreaking via Multi-Agent Collaboration Framework.

---

### Official Review · Reviewer_eKww · 2026-03-14

**Soundness:** 2
**Presentation:** 2
**Significance:** 3
**Originality:** 3
**Overall Recommendation:** 5
**Confidence:** 3

**Summary:**

This paper presents VERA-V, a new jailbreak attack against VLMs by constructing adversarial images against VLMs. Specifically, VERA-V will fine-tune an attacker LLM to generate a pair of a "visual prompt" and a "textual prompt", where the "visual prompt" will then be used to generate adversarial images via a diffusion model, and the "textual prompt" will then be "embedded" as an image that contains textual content. The two images constructed from the prompts will eventually be combined with a set of "distraction images" to form the final "composited" adversarial image. This composited adversarial image can induce targeted victim models to generate targeted harmful content. Experiments are conducted to evaluate the attack performance of VERA-V against real-world models, including both white-box attacks and (transfer-based) black-box attacks.

**Compliance With Llm Reviewing Policy:**

Affirmed.

**Final Justification:**

Thanks to the authors for their rebuttal, which addressed all of my concerns. I have raised my score to 5 to indicate my support.

**Key Questions For Authors:**

See **Strengths and Weaknesses** for details.

**Limitations:**

yes

**Strengths And Weaknesses:**

**Strengths**

1. I think the idea of using (attacker) LLM as a prompt generator to synthesize adversarial images is very interesting and novel. Not like traditional PGD-based adversarial attacks, generating adversarial images via generated prompts and diffusion models can make the final generated images be more robust to various detection defenses, as images synthesisze via diffusion models usually looks natually and does not contain pixel-wise perturbations (which is easy to be detected or distorted).

2. According to Table 1 and Table 2, the proposed VERA-V method significantly surpress other baselines in jailbreaking VLMs, especially black-box VLMs. (However, as I am not an expert in VLMs jailbreaking, I might change my mind if other reviewers show evidence that the performance of VERA-V is not so strong enough)

3. In the experiments, it is good to see results of not only white-box attacks but also (transfer-based) black-box attacks. All these make the propose VERA-V attack more realistic.


**Weaknesses & Questions**

My main concern is that many important ablation studies are missing in the current paper. Specifically:

1. The motivation for using typographic images (with the generated textual prompts) is unclear. Why can one not just simply use the image generated from diffusion models alone to perform the attack? I think the authors should discuss the motivation behind this more, and also perform an ablation study on the role of the typographic image in the final attack performance.

2. In Algorithm 1, is it the case that the hyperparameter $B$ is exactly the term $N$ in Eq.(8)? Besides, it seems that the specific value for the term $N$ is not mentioned in the main paper. I think the authors should clarify what $N$ is used to conduct their main experiments and also add an ablation study on how different values of $N$ would affect the final attack performance.

3. I would also like to see what the performance would be for VERA-V without using the (existing) visual distraction strategy.

Other comments are as follows:

4. For the prior distribution in Eq.(6), what distribution is used in the experiments? Besides, will different priors affect the final attack performance of VERA-V?

5. **Typo:** I think Eq.(3) should be $y \sim P_{VLM}(y \in \mathcal Y_{h} | (x,v))$, right?

---

> ### Author Rebuttal · Authors · 2026-03-31
>
> # Response to Reviewer eKww
>
> We thank the reviewer for the helpful feedback and incorporate the suggested improvements below.
>
>
> ## W1: Ablation Study on Components
>
> We provide this ablation in Appendix E.1 (Table 5), which we expand here with additional configurations:
>
> | Method | ASR | Time Budget |
> |-|-|-|
> | Diffusion only | 12% | 2h 2min |
> | Typography only | 60% | 1h 58min |
> | Distractors only | 0% | 1h 6min |
> | Diffusion + distractors | 34% | 3h 11min |
> | Diffusion + typography | 65% | 2h 26min |
> | Typography + Distractors | 70% | 2h 5min |
> | VERA-V | 80% | 2h 30min |
>
>
> This shows us:
>
> - **Diffusion alone is insufficient:** 12% ASR, implicit visual cues cannot drive attacks without explicit instructions. Adding distractors improves to 34%, but remains weak.
> - **Typography is necessary:** 56% ASR, providing explicit instructions that guide VLM behavior.
> - **Distractors alone have no effect:** 0% ASR, serve only to fragment attention rather than carry adversarial signals.
> - **Cross-modal synergy is critical:** combining diffusion and typography (65%) outperforms either alone, as explicit instructions (typography) are reinforced by implicit contextual cues (diffusion).
>
> Additionally, typography-only variants achieve 56-70% ASR but with higher toxicity rates (29-34%, Table 5), making them easier to detect. VERA-V's full composition achieves best performance (80% ASR) with lowest toxicity (27%) by combining explicit instruction , implicit reinforcement, and attention fragmentation.
>
>
> ## W2:
>
> Yes, $B$ in Algorithm 1 corresponds to $N$ in Eq.(8), both represent the batch size for the Monte Carlo gradient estimate. We apologize for the notation inconsistency and will unify this in the camera-ready revision. We use $B = 8$ for all main experiments, as stated in Appendix C.
>
> We also present an ablation study evaluating the impact of the batch size $N$ on the ASR, using Qwen2.5-VL as the target model. To ensure fair comparison, we fix the total query budget to 40 prompts, which constrains the maximum optimization steps to $S = \lfloor 40 / N \rfloor$.
>
>
> | $N$ value | ASR |
> |-|-|
> | 2 | 36% |
> | 4 | 72% |
> | 8 | 80% |
>
> **Key Insights:** Smaller batch sizes yield suboptimal ASR despite more optimization steps. This is because Monte Carlo estimation suffers from high variance with small batch sizes. Larger batch sizes enable broader exploration of the adversarial prompt space per iteration, allowing Monte Carlo sampling to more reliably steer optimization toward high-reward regions. This yields more robust gradient estimates and a more effective approximation of the target posterior. We cap our evaluation at $N = 8$, which is due to memory constraints of the GPU.
>
>
> ## W3:
>
> Ablation without visual distraction (**Appendix E.1, Table 5**), shows that without distractors,  only diffusion + typography, achieves 65% ASR but with a 68% toxicity rate: meaning the attacks succeed, but will be detectable under external toxicity filters. With distractors, ASR increases to 80% while toxicity drops to 27%, making attacks robust to defense mechanisms. Distractors fragment VLM attention and obscure harmful content, improving stealth without sacrificing effectiveness. Appendix K (Figure 6) provides attention visualizations confirming this fragmentation effect.
>
> ## Q1:
> The prior distribution $P(x_t, x_v)$ in Eq. (6) is the frozen, pre-trained base language model used as the attacker (e.g., Vicuna-7B). During optimization, $q_\theta$ (attacker distribution $q_\theta$ = frozen backbone + LoRA) is updated via LoRA to maximize the attack success rate, while $P$ remains fixed, ensuring stable KL computation and regularizing $q_\theta$ toward fluent, plausible prompts.
>
> Changing the prior is equivalent to changing the attacker's frozen backbone. Appendix E.4 (Table 8) shows results across three backbones: Vicuna-7B (80% ASR), Mistral-7B (76%), and LLaMA3-8B (70%). VERA-V remains effective across all choices, demonstrating robustness to the prior.
>
> ## Q2:
>
> Eq.(3) is intentionally written this way to frame jailbreak generation as posterior inference: we want to sample adversarial input pairs $(x,v)$ from the posterior distribution conditioned on the VLM producing harmful outputs ($y \in Y_h$, denoted as $y^\*$). Thus, $(x,v) \sim P_{\text{VLM}}((x,v) | y^{*})$ represents the posterior distribution over adversarial inputs that induce harmful behavior. This posterior perspective is what enables VERA-V to:
>
> 1. Generate diverse jailbreaks (sampling from a distribution, not finding one point)
> 2. Use variational inference to approximate this intractable posterior with $q_\theta(x_t, x_v)$
> 3. Derive the ELBO objective in Eq.(6) via Bayes' rule (Eq.(13) in Appendix B.1)
>
> The forward likelihood $P_{\text{VLM}}(y|x,v)$ appears in Eq.(6) after applying Bayes' rule.

---

> > ### Author Rebuttal · Reviewer_eKww · 2026-04-03
> >
> > Thanks to the authors for their rebuttal, which addressed all of my concerns. I have raised my score to 5 to indicate my support.

---

> > > ### Author Response · Authors · 2026-04-05
> > >
> > > # Final reply to reviewer eKww
> > >
> > > We thank the reviewer for the constructive feedback and for the time dedicated to reviewing our work!

---

### Decision · Program_Chairs · 2026-04-30

**Decision:**

Accept (regular)

**Comment:**

This paper introduces VERA-V, a variational inference framework that recasts multimodal jailbreak discovery as learning a joint posterior distribution over paired text-image prompts. This probabilistic view captures complex cross-modal interactions, enabling stealthy, coordinated adversarial inputs that bypass model guardrails.

After rebuttal, this paper has mixed scores with 5,5,3,3. Before rebuttal, almost all reviewers point out the problem of ablations on different components of VERA-V. After rebuttal, main concerns from reviewers are 1) whether VERA-V exploits cross-modal coupling or mainly behaves as a stronger optimized typography attack with auxiliary visual components and 2) some presentation concerns.  AC feels presentation concerns can be solved during the rebuttal.  AC suggests the weak accept for this paper.